# The Anti-Inflammatory Effect of Hydrogen Gas Inhalation and Its Influence on Laser-Induced Choroidal Neovascularization in a Mouse Model of Neovascular Age-Related Macular Degeneration

**DOI:** 10.3390/ijms222112049

**Published:** 2021-11-07

**Authors:** I-Chia Liang, Wen-Chin Ko, Yu-Jou Hsu, Yi-Ru Lin, Yun-Hsiang Chang, Xv-Hui Zong, Pei-Chen Lai, Der-Chen Chang, Chi-Feng Hung

**Affiliations:** 1Department of Ophthalmology, Tri-Service General Hospital, National Defense Medical Center, Taipei 11490, Taiwan; ysonyaliang@gmail.com (I.-C.L.); yun.siang@me.com (Y.-H.C.); 2Ph.D. Program in Nutrition and Food Science, Fu Jen University, New Taipei City 24205, Taiwan; 3School of Medicine, Fu Jen Catholic University, New Taipei City 24205, Taiwan; 086938@mail.fju.edu.tw; 4Division of Cardiac Electrophysiology, Department of Cardiovascular Center, Cathay General Hospital, Taipei 10630, Taiwan; 5Graduate Institute of Biomedical and Pharmaceutical Science, Fu Jen Catholic University, New Taipei City 24205, Taiwan; s16179263@gmail.com; 6Department of Ophthalmology, Cathay General Hospital, Taipei 10630, Taiwan; yirulin83088@gmail.com; 7Tsung Cho Chang Laboratory, College of Medicine, Fu-Jen Catholic University, New Taipei City 24205, Taiwan; 054317@mail.fju.edu.tw; 8Institute of Biochemistry and Molecular Biology, College of Medicine, National Taiwan University, Taipei 100233, Taiwan; peggylai1116@gmail.com; 9Department of Mathematics and Statistics and Department of Computer Science, Georgetown University, Washington, DC 20057, USA; chang@georgetown.edu; 10School of Pharmacy, Kaohsiung Medical University, Kaohsiung 80708, Taiwan

**Keywords:** age-related macular degeneration, choroidal neovascularization, hydrogen gas

## Abstract

Background: Age-related macular degeneration (AMD) is a leading cause of blindness in the elderly. Choroidal neovascularization (CNV) is the major pathologic feature of neovascular AMD. Oxidative damages and the ensuing chronic inflammation are representative of trigger events. Hydrogen gas (H_2_) has been demonstrated as an antioxidant and plays a role in the regulation of oxidative stress and inflammation. This experiment aimed to investigate the influence of H_2_ inhalation on a mouse model of CNV. Methods: Laser was used to induce CNV formation. C57BL/6J mice were divided into five groups: the control group; the laser-only group; and the 2 h, 5 h, and 2.5 h/2.5 h groups that received laser and H_2_ inhalation (21% oxygen, 42% hydrogen, and 37% nitrogen mixture) for 2 h, 5 h, and 2.5 h twice every day, respectively. Results: The severity of CNV leakage on fluorescence angiography showed a significant decrease in the H_2_ inhalation groups. The mRNA expression of hypoxia-inducible factor 1 alpha and its immediate downstream target vascular endothelial growth factor (VEGF) showed significant elevation after laser, and this elevation was suppressed in the H_2_ inhalation groups in an inhalation period length-related manner. The mRNA expression of cytokines, including tumor necrosis factor alpha and interlukin-6, also represented similar results. Conclusion: H_2_ inhalation could alleviate CNV leakage in a laser-induced mouse CNV model, and the potential mechanism might be related to the suppression of the inflammatory process and VEGF-driven CNV formation.

## 1. Introduction

Age-related macular degeneration (AMD), the leading cause of blindness in elderly people aged over 60, is estimated to have an increasing prevalence globally as a consequence of exponential population growth, increase in life expectancy, and falling death rates [1,2,3]. The prevalence of AMD increases exponentially with rising age, which does not differ between males and females [2,4,5]. Across racial populations, there is a higher prevalence of AMD in Europeans than in Asians and Africans, while there is no difference between Asians and Africans. Europeans have a higher prevalence of the geographic atrophy subtype than Africans, Asians, and Hispanics. Between geographical regions, AMD was less prevalent in Asia than in Europe and North America [2]. Increasing age and genetic factors play important roles in the development of AMD, while other risk factors, including smoking, obesity, a higher body mass index, and metabolic syndromes, such as dyslipidemia, hypertension, and hyperglycemia, are reported to be associated with higher prevalence, more progression, and increased severity of AMD [1,2,6,7,8]. AMD stems from accumulation of drusen, the lipoprotein-rich deposits underneath retinal pigment epithelium (RPE) with pigmentary changes and/ or thickened overlying Bruch’s membrane. Progressive drusen accumulation results in further inflammation and alterations of RPE as the disease progresses. Advanced AMD is classified to be of the dry (atrophic) or wet (neovascular) type. Advanced dry AMD is characterized by progressive atrophic changes in the RPE and overlying neurosensory retina with an eventual development of geographic atrophy. Proliferation of choroidal neovascularization (CNV) underneath the neurosensory retina contributes to wet AMD [7,9]. Micronutrient supplements may be beneficial for intermediate and advanced AMD. The AREDS study showed a reduction in the progression from intermediate AMD to advanced AMD with a daily formula of 500 mg vitamin C, 80 mg zinc, 400 IU vitamin E, 2 mg copper, and 15 mg beta carotene. A similar effect was observed in the AREDS2 study, with a replacement of beta carotene with 10 mg lutein and 2 mg zeaxanthin [10,11]. Lifestyle modification is also recommended to inhibit the progression of AMD [12,13].

Disrupted regulation of angiogenesis is the main pathogenesis of CNV. Vascular endothelial growth factor (VEGF) has been identified as a critical mediator of pathologic neovascularization. The level of VEGF elevates when damage, hypoxia, and ischemia occur. VEGF acts by promoting endothelial cell proliferation and migration, and it is the main regulator of downstream factors. At present, VEGF-targeted treatment is the mainstream treatment for neovascular AMD [9,14]. However, the proceeding inflammation and subsequent atrophic changes also play important roles. Increased VEGF alone may not be enough to bring about CNV development, which implies that the VEGF-driven pathways are only part of the complex machinery regulation [15]. VEGF is constitutively produced in the eye and is essential for normal physiology. The adverse effects related with chronic VEGF suppression occur if the loss of pro-survival and neuroprotective capacities of VEGF happen [16]. VEGF-independent regulation of ocular angiogenesis, including neuron survival promotion or immune regulation, is currently under evaluation and is considered an important alternative treatment [17].

Hydrogen gas, an inert gas, has been reported by Ohsawa et al. in 2007 [18] to be an efficient antioxidant in combating oxidative brain injury when administered by inhalation. This finding raises the interest and attracts the attention of various medical communities. Its benefits in several ophthalmic disease models, including retinal ischemia/reperfusion injury, retinal vein occlusion, and endotoxin-induced uveitis, have been evaluated [19,20,21]. However, its role in a laser-induced AMD animal model has not been evaluated yet.

## 2. Results

### 2.1. Color Fundus Photography (CFP) and Fluorescence Angiography (FA) Analysis of the Effects of Hydrogen Gas Inhalation on Laser-Induced CNV (n = 3)

Four spots of laser photocoagulation per eye were performed to induce CNV formation in C57BL/6J mice on Day 5. Hydrogen inhalation was given since Day 1 as pre-treatment and was continued after laser until Day 15, the end of the experiment. CFP and FA results on both Day 10 and Day 15 are shown in Figure 1a and a. FA leakage scores for each of the four laser spots in each eye were plotted and calculated for all groups. The results of the FA leakage scores on both Day 10 and Day 15 are shown in Figure 1b and Figure 2b and reveal that hydrogen inhalation led to a significant reduction in the FA leakage scores when compared to those without hydrogen inhalation. Moreover, the effect was inhalation time related.

### 2.2. Histology, Immunofluorescence Staining, and the Effect of Hydrogen as Inhalation on the Reduction in Phosphorylation of the VEGF Receptor

Hematoxylin–eosin (HE) staining of the CNV is shown in Figure 3 and reveals almost no RPE coverage over the CNV tissue in the laser-only group and more RPE coverage over the CNV tissue after hydrogen inhalation. A longer inhalation time resulted in more RPE coverage. The CNV tissue was revealed to be spindle-shaped in the laser-only group and became flattened after hydrogen inhalation.

Immunofluorescence staining is showed in Figure 4. The phosphorylation of VEGF receptor (phospho-VEGF receptor, pVEGFR) decreased after hydrogen inhalation.

### 2.3. Hydrogen Gas Inhalation Downregulates the mRNA Expression of Hypoxia-Inducible Factor 1 alpha (HIF-1α) and Its Immediate Downstream Target VEGF in Laser-Induced CNV (n = 3)

To understand the mRNA expression after CNV induction and hydrogen gas inhalation, qRT-PCR was used. The results are shown in Figure 5. The expression of HIF-1α and VEGF increased significantly after laser photocoagulation compared with those of the control group (no laser or hydrogen inhalation). Hydrogen inhalation effectively down-regulated the mRNA expression of HIF-1α and VEGF. The effect was more significant in the 5 h group (a longer inhalation time of 5 h per day) compared to the 2 h group (a shorter inhalation time of 2 h per day). The effect was similar between the 5 h group and the 2.5 h/2.5 h group, which indicated that interrupted inhalation could achieve a similar effect as continuous inhalation.

### 2.4. Hydrogen Gas Inhalation Reduces the mRNA Levels of Tumor Necrosis Factor Alpha (TNF-α) and Interlukin-6 (IL-6) (n = 3)

The mRNA expression of pro-inflammatory mediators-TNF-α and IL-6 were assayed by qRT-PCR and are shown in Figure 6. The expression of TNF-α and IL-6 revealed significant elevation after laser photocoagulation compared with those of the control group (no laser or hydrogen inhalation). This elevation could be effectively downregulated by hydrogen inhalation. The effects of the 5 h and 2.5 h/2.5 h groups were about the same and were better than those of the 2 h group.

## 3. Discussion

This study showed that hydrogen gas administered via inhalation could protect the sensory retina after the injury of Bruch’s membrane disruption, and the following CNV formation was revealed. We found that hydrogen gas could inhibit the inflammation of an injured retina after Bruch’s membrane disruption, suppress CNV development, and decrease the leakage of the CNV.

AMD is regarded as a disease of complex multifactorial mechanisms, with aging being one of the main risk factors. While risk factors such as aging and genetic susceptibility are strong and non-modifiable, other risk factors, including diet, smoking, and metabolic disturbances, are modifiable to some extent [22]. Neovascular AMD, predisposed by a dry AMD with continuous progression of RPE atrophy, is characterized by the formation of CNV.

We demonstrated that the leakage of CNV could be significantly decreased by hydrogen gas inhalation. The effect showed inhalation time dependency.

The histological examination of surgically excised CNV from AMD patients showed that the CNV is composed of the vascular component (endothelial cells, pericytes, and precursors of endothelial cells) and the extravascular component (glial cells, RPE cells, fibroblasts, and inflammatory cells, including macrophages, lymphocytes, granulocytes, and foreign body giant cells) [23,24,25]. Consequently, both inflammation and angiogenesis are involved in the development of CNV.

VEGF has been identified as one of the critical mediators of pathologic neovascularization and its role in ocular angiogenesis has been evaluated in detail [14,26]. VEGF is essential in the initial development of CNV, as the inciting stimulus involved in the proliferation and migration of endothelial cells. RPE cells, ganglion cells, retinal neurons, Müller cells, and immune cells, such as macrophages and monocytes, secrete VEGF in response to hypoxic and inflammatory stimuli [26,27,28,29]. The response to hypoxia is mediated by HIF, a family of transcription regulators that regulates the genes involved in the response to hypoxia, including VEGF. HIF-1α is the inducible subunit of the HIF-1 transcription factor and has been discovered to be present in active CNV specimens. It plays an important role in hypoxia signaling during the development and progression of vision-threatening production of CNV [30,31].

At present, based on the ideas of VEGF as the main driver of angiogenesis in CNV, the gold standard therapy for neovascular AMD involves intravitreal administration of VEGF inhibitors. The agents used clinically include bevacizumab, ranibizumab, aflibercept, and brolucizumab. In our present experiment, the mRNA expression of HIF-1α and VEGF could be successfully inhibited by hydrogen gas inhalation in an inhalation time-related manner. Hydrogen gas inhalation has been shown to alleviate retinal edema and promote retinal function recovery in branched retinal vein occlusion rat model via reducing VEGF expression [19]. Our results revealed that not just VEGF but the upstream translation factor HIF could also be inhibited by hydrogen gas inhalation.

Anti-VEGF pharmacotherapy could successfully inhibit CNV activity and achieve resolution of leakage of blood vessels and hemorrhage in the macula associated with CNV but is not enough to eradicate CNV and cure the disease. CNV persists with only partial regression or recurs years later [17,32]. Moreover, a subset of patients was found to be non-responsive to anti-VEGF therapy [33,34]. As such, VEGF-driven pathways are only a part of the complex machinery regulating angiogenesis in the eye. Regulating ocular angiogenesis other than VEGF is the current goal in the treatment of neovascular AMD.

Macrophages and numerous cytokines have been known to be important players in regulating ocular angiogenesis through multiple mechanisms. Macrophages are critical for the proper retinal vascular remodeling and also play pivotal roles in modulating retinal neovascularization [17,35,36]. Depletion of macrophage activity could decrease both the size and leakage of CNV [37]. Leukocytes would be recruited under hypoxia and the increasing macrophages in broken Bruch’s membrane elicit the production of TNF-α and the synthesis of ILs as well as stimulate chemokines production from RPE cells [38]. TNF-α was found to be the major cytokine responsible for the macrophage-derived angiogenesis [39]. A nearly five-fold increase in the prevalence of neovascular AMD was noted in patients with blood monocytes expressing higher TNF-α mRNA levels [40]. TNF receptor inhibitor was found to be able to inhibit the blood–retinal barrier breakdown in rat models of inflammation and diabetic retinopathy [41,42]. Angiogenesis-related effects and induction of ocular inflammation are two of the most important effects of IL-6 in the eye [43]. High levels of IL-6 in the blood of patients with neovascular AMD are shown to be positively correlated with disease progression [44]. Despite playing roles in the initial stage of CNV, IL-6 expressed by activated macrophages has been noted to be increased in the CNV lesion of neovascular AMD and that the inhibition of IL-6 signaling can suppress subretinal fibrosis [45,46], which is an important manifestation of end-stage AMD. Targeting IL-6 and the corresponding signaling pathway would bring us close to not only treating CNV but also preventing subretinal fibrosis and eventual scarring of the macula.

Hydrogen gas inhalation has been proven to decrease the levels of inflammatory cytokines, including TNF-α, IL-1β, and IL-6, in ischemia/reperfusion injury or the retina, liver, and brain [20,47,48]. In our current experiment, the gene expression of TNF-α and IL-6 could be effectively suppressed by hydrogen inhalation. An inhalation time-dependent effect was also revealed.

The main finding of the current experiment demonstrated that hydrogen inhalation could (1) decrease the leakage of CNV; (2) suppress the VEGF-dependent pathway of CNV formation via downregulation of VEGF and HIF-1α; and (3) inhibit the VEGF-independent pathway via restraining the expression of TNF-α and IL-6. Moreover, the observation that the longer the time of the treatment is, the better the effect is, confirmed an inhalation time-related manner.

## 4. Materials and Methods

### 4.1. Animals

Four to six weeks of age C57BL/6J mice were obtained from the National Biotechnology Research Park, Taipei, Taiwan.

The mice were divided into five groups as follows: (1) the control group received neither laser nor hydrogen gas inhalation; (2) the laser-only group received laser but no hydrogen gas inhalation; (3) the 2 h group received laser and a continuous 2 h hydrogen gas inhalation every day; (4) the 5 h group received laser and a continuous 5 h hydrogen gas inhalation every day; and (5) the 2.5 h/2.5 h group received laser and two periods of 2.5 h hydrogen gas inhalation interrupted by a 3 h rest every day.

Mice were anesthetized with intraperitoneal (IP) injection of Zoletil™50 (zolazepam + tiletamine) and rompun 20 (xyalzine) in a 3:2 mixture, 1 uL/1 g, and the pupils were dilated with 0.5% phenylephrine hydrochloride and 0.5% tropicamide (Santen Pharmaceutical Co., Ltd., Tokyo, Japan) during laser treatment, CFP, and FA

### 4.2. Hydrogen Inhalation

A transparent airtight acrylic box (20 × 18 × 15 cm, length × width × height) with an outflow only fan on the top was set as the inhalation chamber (Figure 7). Hydrogen and an oxygen mixture gas (67% hydrogen mixed with 33% oxygen) produced from deionized water by electrolysis with a hydrogen/oxygen generator (model AMS-H-03; Asclepius Meditec Co., Ltd., Shanghai, China) was directly introduced into the inhalation chamber at a rate of 3 L/min. Nitrogen gas was also applied and adjusted the oxygen concentration to 21%. The inhalation chamber was flushed 30 min in advance with experimental gas to replace the room air inside. Animals needing hydrogen inhalation (group 3, 4, and 5) received the mixture of 21% oxygen, 42% hydrogen, and 37% nitrogen gas, according to different group protocols since Day 1 as pretreatment. The inhalation was enhanced to a continuous 8 h on Day 5 just after laser treatment in all three groups (3, 4, and 5). The hydrogen inhalation was then set back to initial protocols since Day 6 and continued after Day 15, the end of the experiment.

### 4.3. Laser-Induced CNV Model

Laser photocoagulation was performed on Day 5 to induce CNV as previously reported [49,50]. Briefly, mice were anesthetized, and the pupils were dilated in advance. Laser photocoagulation spots (100 um size, 0.15 s duration, 150 mW) were made with 532 nm green laser photocoagulator (LIGHTLas 532, LIGHTMED, San Clemente, CA, USA) on each retina with a slit-lamp delivery system and a hand-held coverslip as a contact lens. Four burns were made at 3, 6, 9, and 12 o’clock positions around the optic nerve head. A production of a bubble at the time of laser irradiation, which indicated a rupture of the Bruch’s membrane, suggested the effective point that induced a CNV.

### 4.4. Color Fundus Photography and Fluorescene Angiography

A Micron IV retinal imaging microscope (Phoenix Research Laboratories, Pleasanton, CA, USA) was used to monitor morphological and pathological changes in the fundus of C57BL/6J mice. Briefly, mice were anesthetized, and the pupils were dilated in advance. Each mouse was held on its side on the microscope platform and the eye was rinsed with 2% Methocel gel (OmniVision, SA, Neuhausen, Switzerland). After CFP was performed, fluorescein (10%; 0.05 mL) was used for FA examination through IP injection. Serial images and videos were then collected.

Two retinal specialists (Liang and Chang) evaluated the angiograms for FA grading evaluation in a blinded manner using a grading system [51], where Grade 1 = no hyperfluorescence; Grade 2 = hyperfluorescence without leakage; Grade 3 = hyperfluorescence in the early or middle phase and leakage in the late phase; and Grade 4 = bright hyperfluorescence in the transit and leakage in the late phase beyond the treated areas. Subretinal hemorrhages seldom occurred during laser photocoagulation and resulted in fluorescein blockage during FA. This situation would influence the evaluation and these laser spots were excluded from FA grading evaluation to avoid significant bias. Total grades were analyzed for statistical significance.

### 4.5. Histology and Immunofluorescence Staining

Mice were euthanized on Day 15. Both eyes of each mouse were enucleated and rapidly frozen in embedding medium prior to sectioning. The optic nerve parallel to the sagittal plane at the laser photocoagulation position was selected, and slices with a thickness of 4.0 um were prepared continuously. The sections were stained with HE, observed, and photographed using a light microscope (EVOS M7000 Imaging System, Thermo Fisher, Waltham, MA, USA).

Cryosections for immunofluorescence staining were thawed, air-dried, and fixed in 10% paraformaldehyde with 0.3% TritonX-100 at room temperature for 1 h. Immuno-fluorescence staining was performed according to the manufacturer’s instructions. Next, the sections were washed in PBS 3 times for 5 min. Then, the sections were incubated with pVEGFR primary antibodies (#3817, Cell Signaling, Danvers, MA, USA) at a 1:100 dilution overnight at 4 °C. The slides were washed with PBS 3 times for 5 min and incubated with an Alexa Fluor 594-conjugated goat anti-rabbit IgG (H + L) secondary antibodies (#8889, Cell Signaling, Danvers, MA, USA) at 1:500 dilution for 2 h at room temperature. The nuclei were stained with 100 ng/mL DAPI for 10 min. The sections were washed with PBS between these incubations. Finally, the sections were mounted and examined and captured with a fluorescence microscope (EVOS M7000 Imaging System, Thermo Fisher, Waltham, MA, USA).

### 4.6. Quantitative Real-Time Polymerase Chain Reaction (qRT-PCR) to Measure Transcription Levels

Animals were sacrificed on day 15 and both eyes of each mouse were enucleated for qRT-PCR. Total RNA of each eye was isolated using the total RNA Isolation kit (GeneDireX, Inc., Taiwan) according to the manufacturer’s instructions and reverse-transcribed into cDNA using iScript. The qPCR was performed using the StepOnePlus. Real-Time PCR System (Applied Biosystems, Foster City, CA, US) with SYBR green (Applied Biosystems, Foster City, CA, US). Primers for determining the expression of genes encoding hypoxia-inducible factor 1 alpha (HIF-1α), VEGF, tumor necrosis factor alpha (TNF-α), and interlukin-6 (IL-6) are listed in Table 1.

### 4.7. Statistical Analysis

Statistically significant differences between groups were determined using one-way analysis of variance (ANOVA) with MedCalc software. A *p*-value < 0.05 was considered statistically significant.

## 5. Conclusions

In summary, our study revealed the bio-characteristics of hydrogen gas and its influences in CNV. The effect was achieved by suppression of both the VEGF pathway and inflammatory chemokines.

Anti-VEGF is the main clinical treatment for neovascular AMD at present, despite cases of non-responsiveness and disease recurrences remaining unsolved problems. The present study provided evidence for hydrogen inhalation as an ancillary candidate in preventing and modulating neovascular AMD via multiple pathways.

## Figures and Tables

**Figure 1 ijms-22-12049-f001:**
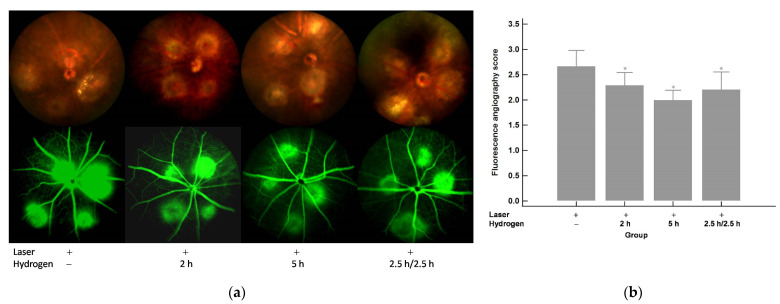
Day 10 (5 days after laser photocoagulation). (**a**) CFP and FA. (**b**) FA leakage scores. Significantly lower FA leakage scores were shown after the hydrogen inhalation compared to those without inhalation and the effect was inhalation period related. * *p* < 0.05 compared with the laser+/hydrogen− group.

**Figure 2 ijms-22-12049-f002:**
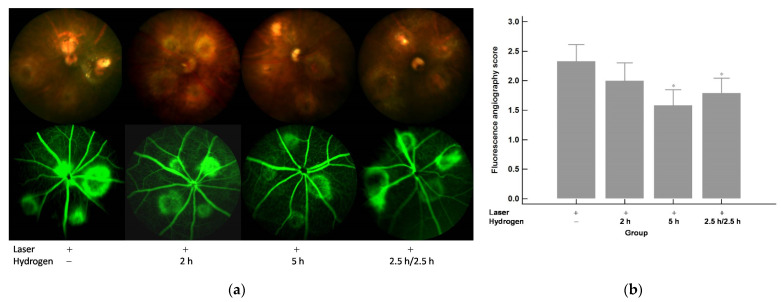
Day 15 (10 days after laser photocoagulation). (**a**) CFP and FA. (**b**) FA leakage scores. Significantly lower FA leakage scores were shown after the hydrogen inhalation compared to those without inhalation and the effect was inhalation period related. * *p* < 0.05 compared with the laser+/hydrogen− group.

**Figure 3 ijms-22-12049-f003:**
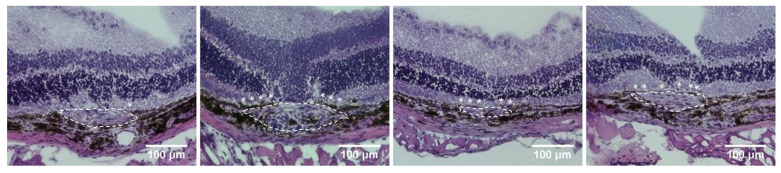
H&E staining on Day 15 (10 days after laser photocoagulation). Left to right: Laser-only group, 2 h group, 5 h group, and 2.5 h/2.5 h group. The shape of the CNV (white dotted line) became flattened after hydrogen inhalation. More RPE coverage (dark-pigmented tissue labeled with an asterisk) over the CNV tissue was also noted after hydrogen inhalation.

**Figure 4 ijms-22-12049-f004:**
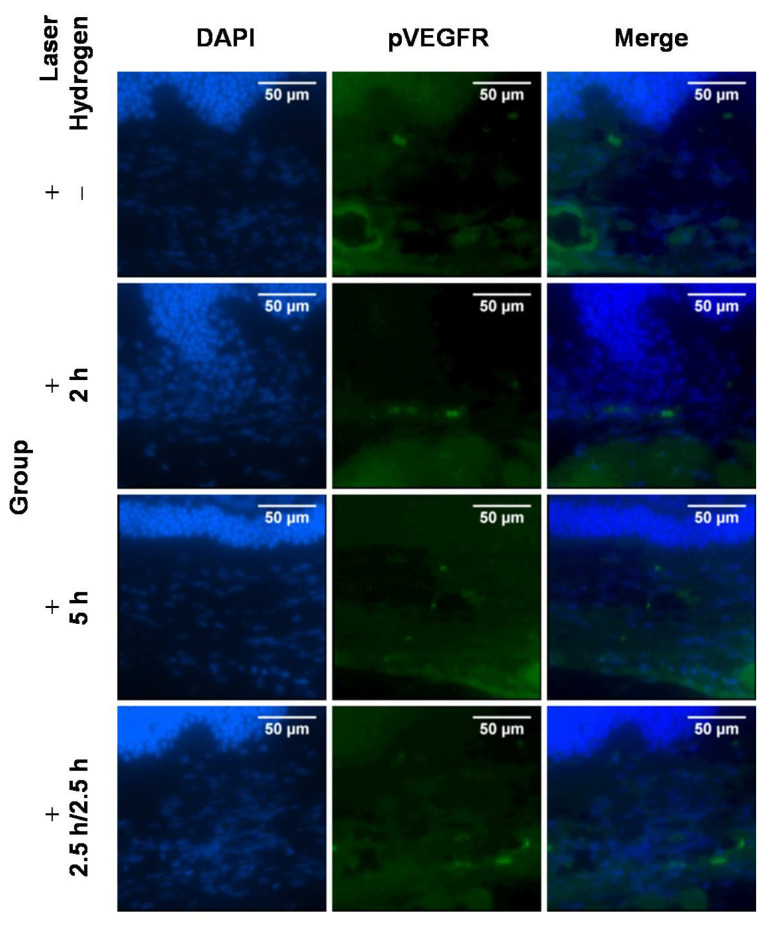
Immunofluorescence staining and the expression of pVEGFR on Day 15 (10 days after laser photocoagulation).

**Figure 5 ijms-22-12049-f005:**
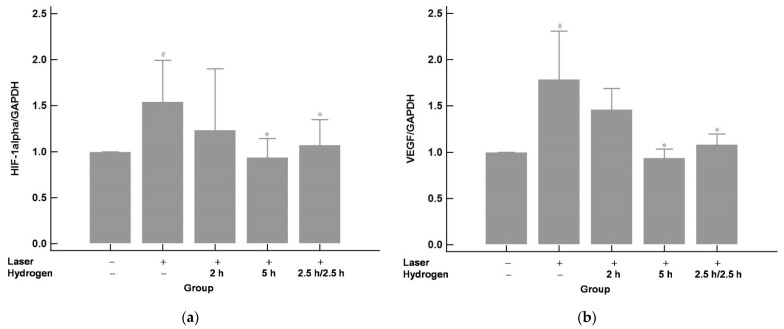
The mRNA expression of HIF-1α (**a**) and VEGF (**b**). The expression of HIF-1α and VEGF increased after laser photocoagulation compared with those of the control (no laser or hydrogen inhalation). The elevated HIF-1α and VEGF expression was downregulated by hydrogen inhalation. The effect was similar between the 2.5 h/2.5 h and 5 h groups and showed to be less effective in the 2 h group. # *p* < 0.05 compared with the control (laser−/hydrogen−) group; * *p* < 0.05 compared with the laser+/hydrogen− group.

**Figure 6 ijms-22-12049-f006:**
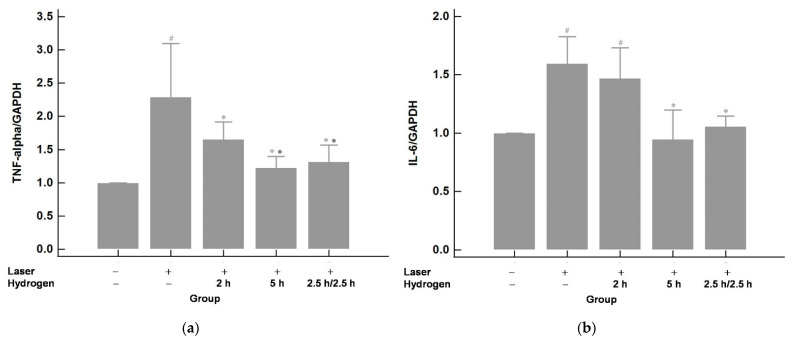
The mRNA expression of TNF-α (**a**) and IL-6 (**b**). The mRNA levels of TNF-α and IL-6 were upregulated after laser photocoagulation compared with those of the control (no laser or hydrogen inhalation). The upregulation could be suppressed by hydrogen inhalation. The effects of the 2.5 h/2.5 h and 5 h groups were similar and were more effective than the 2 h group. # *p* < 0.05 compared with the control (laser−/hydrogen−) group; * *p* < 0.05 compared with the laser+/hydrogen− group; ● *p* < 0.05 compared with the laser+/hydrogen 2 h group.

**Figure 7 ijms-22-12049-f007:**
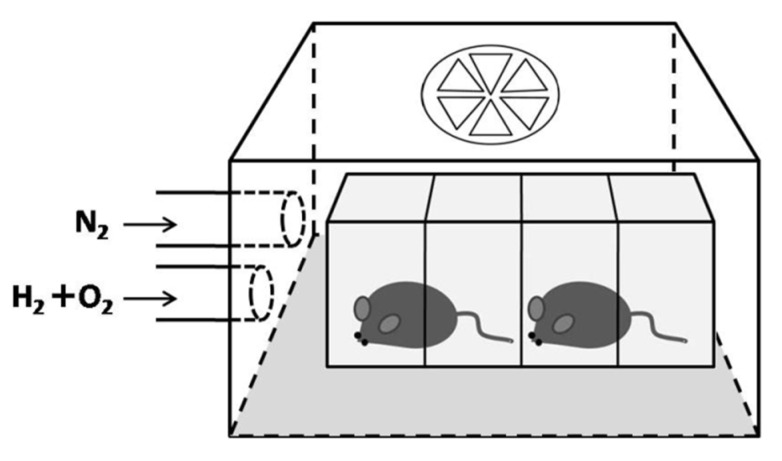
The setting of the inhalation chamber.

**Table 1 ijms-22-12049-t001:** Primers of reverse transcription PCR analysis for genes.

Primer	Primer Sequence (5′ -3′)	Primer Sequence (3′ -5′)	Product Length (bp)
HIF-1α	CCAGCAGACCCAGTTACAGA	TGAGTGCCACTGTATGCTGA	20
VEGF	CTGCTGTAACGATGAAGCCCTG	GCTGTAGGAAGCTCATCTCTCC	22
TNF-α	GGTGCCTATGTCTCAGCCTCTTTT	GCCATAGAACTGATGAGAGGGAG	23
IL-6	AGTTGCCTTCTTGGGACTGA	TCCACGATTTCCCAGAGAAC	20

## Data Availability

Not applicable.

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
