# Peer review of "The Anti-Inflammatory Effect of Hydrogen Gas Inhalation and Its Influence on Laser-Induced Choroidal Neovascularization in a Mouse Model of Neovascular Age-Related Macular Degeneration"

_ijms, 2021, doi:10.3390/ijms222112049_

Round 1

Reviewer 1 Report

Introduction section

There was no mention of the sex difference in the prevalence of AMD across populations.

Result section

Grammatical error in line 102 “onbothday10” should be written as “on both day 10.”

Section 2.4

Grammatical error in line 148 “therapyreduces” should be written as “therapy reduces.”

Reviewer 2 Report

The authors have presented data that hydrogen inhalation therapy decreased inflammation (TNF-a, IL-6) and mechanisms of injury (VEGF) associated with macular degeneration in a rodent model of laser induced retinal injury. The data in figures 1, 2, 5 and 6 strongly support their assertions that hydrogen gas inhalation improved healing in this model. However, figure 4 fails to demonstrate clear changes in pVEGF without quantitation or better images, and figure 3 is difficult for some readers to interpret without prior anatomical expertise in the retina - please label the RPE and CNV discussed in the legend and text so readers will see what the authors claim to see. It is hard for single images to convince an audience. 

Major concerns:

  1. The premise is that hydrogen inhalation therapy could treat AMD, however this is not an age model, though the model is a good inflammation model. Are the same effects seen if the animals are not pretreated prior to injury? That is, can the therapy begin 1 or 5 days post injury and still have a beneficial effect? If the point of the manuscript is a therapy, that becomes a critical time frame to understand. If the point of the manuscript is that hydrogen inhalation therapy is anti-inflammatory and that inflammation and VEGF drive macular degeneration in this model, then the pretreatment is better understood.

Minor concerns:

  1. Label figure 3 for histology please.
  2. Check verb and noun tenses in the English translations. 
  3. The word tendency is used to describe graphs that show asterisks for significance - please clarify/consider your use of 'tendency'
  4. title 4.6 repeats the first word
